Swarm: robust and fast clustering method for amplicon-based studies

Mahé Frédéric 1 2 3 mahe@rhrk.uni-kl.de
Rognes Torbjørn 4 5
Quince Christopher 6
de Vargas Colomban 1 2
Dunthorn Micah 3
1 CNRS, UMR 7144, EPEP – Évolution des Protistes et des Écosystèmes Pélagiques, Station Biologique de Roscoff , Roscoff , France
2 Sorbonne Universités, UPMC Univ Paris 06, UMR 7144, Station Biologique de Roscoff , Roscoff , France
3 Department of Ecology, University of Kaiserslautern , Kaiserslautern , Germany
4 Department of Microbiology, Oslo University Hospital, Rikshospitalet , Oslo , Norway
5 Department of Informatics, University of Oslo , Oslo , Norway
6 School of Engineering, University of Glasgow , Glasgow , UK
Cohan Frederick
Electronic publication date: 2014 Sep 25
Publication date: 2014
Volume: 2
Electronic Location ID: e593
Received 2014 May 13; Accepted 2014 Sep 3
Copyright: © 2014 Mahé et al.
Copyright year: 2014
Copyright holder: Mahé et al.
License: This is an open access article distributed under the terms of the Creative Commons Attribution License, which permits unrestricted use, distribution, reproduction and adaptation in any medium and for any purpose provided that it is properly attributed. For attribution, the original author(s), title, publication source (PeerJ) and either DOI or URL of the article must be cited.
License URL: https://creativecommons.org/licenses/by/4.0/

Keywords: Environmental diversity, Barcoding, Molecular operational taxonomic units

Funding: EU EraNet BiodivErsA program BioMarKs 2008-6530 French government “Investissements d’Avenir” project OCEANOMICS ANR-11-BTBR-0008 Deutsche Forschungsgemeinschaft DU1319/1-1 EPSRC Career Acceleration Fellowship EP/H003851/1 FM and CdeV were supported by the EU EraNet BiodivErsA program BioMarKs (grant #2008-6530) and the French government “Investissements d’Avenir” project OCEANOMICS (ANR-11-BTBR-0008) and the EU FP7 program MicroB3 (contract number 287589). FM and MD were supported by the Deutsche Forschungsgemeinschaft (grant #DU1319/1-1). TR was supported by a Centre of Excellence grant from the Research Council of Norway to CMBN. CQ is funded by an EPSRC Career Acceleration Fellowship – EP/H003851/1. The funders had no role in study design, data collection and analysis, decision to publish, or preparation of the manuscript.

==============================
Popular de novo amplicon clustering methods suffer from two fundamental flaws: arbitrary global clustering thresholds, and input-order dependency induced by centroid selection. Swarm was developed to address these issues by first clustering nearly identical amplicons iteratively using a local threshold, and then by using clusters’ internal structure and amplicon abundances to refine its results. This fast, scalable, and input-order independent approach reduces the influence of clustering parameters and produces robust operational taxonomic units.

Introduction

High-throughput sequencing technologies can generate millions of amplicons (or barcode sequences) in a single run, and are thus today our best approach to deeply assess the environmental or clinical diversity of complex microbial assemblages of archaea, bacteria, and eukaryotes. The millions, and soon billions, of raw reads produced in molecular ecology and metabarcoding projects need to be clustered into molecular operational taxonomic units (OTUs) before being used for diversity estimates or other statistical analyses.

Because of the increasing sizes of today’s amplicon datasets, fast and greedy de novo clustering heuristics are the preferred and only practical approach to produce OTUs (Edgar, 2010; Ghodsi, Liu & Pop, 2011; Fu et al., 2012). Shared steps in these current algorithms are: an amplicon is drawn out of the amplicon pool and becomes the center of a new OTU (centroid selection), this centroid is then compared to all other amplicons remaining in the pool. Amplicons for which the distance is within a global clustering threshold, t, to the centroid are moved from the pool to the OTU. The OTU is then closed. These steps are repeated as long as amplicons remain in the pool (Fig. 1A).

These greedy clustering methods suffer from two fundamental problems. First, they use an arbitrary fixed global clustering threshold. As lineages evolve at variable rates, no single cut-off value can accommodate the entire tree of life. A single global clustering threshold will inevitably be too relaxed for slow-evolving lineages and too stringent for rapidly evolving ones (Stackebrandt & Goebel, 1994; Sogin et al., 2006; Nebel et al., 2011; Koeppel & Wu, 2013). Second, the input order of amplicons strongly influences the clustering results. Previous centroid selections are not re-evaluated as clustering progresses, which can generate inaccurately formed OTUs, where closely related amplicons can be separated and unrelated amplicons can be grouped (Koeppel & Wu, 2013) (Fig. 1A).

Figure 1 Schematic view of the greedy clustering approach and comparison with swarm.

(A) Visualization of the widely used greedy clustering approach based on centroid selection and a global clustering threshold, t, where closely related amplicons can be placed into different OTUs. (B) By contrast, Swarm clusters iteratively by using a small user-chosen local clustering threshold, d, allowing OTUs to reach their natural limits.

Swarm’s rationale

While working on two large scale environmental diversity studies using different markers and different sequencing platforms—the BioMarKs project (e.g., Bittner et al., 2013; Dunthorn et al., 2014; Logares et al., 2014) and the TARA OCEANS project (e.g., Karsenti et al., 2011)—the limitations of greedy de novo clustering methods became salient. We observed that amplicons from one species can be subsumed into the OTU of a genetically closely related species with a very dissimilar ecology if that second species had a higher abundance, leading to erroneous ecological interpretations. To solve these issues, we developed Swarm—a novel method that avoids both fixed global clustering thresholds, and input-order dependency due to centroid selection. Our objective was to implement an exact, yet fast, de novo clustering method that produces meaningful OTUs and reduces the influence of clustering parameters.

Swarm can be defined as a fast and exact, two-phased, agglomerative, unsupervised (de novo) single-linkage-clustering algorithm. During the growth phase, Swarm computes sequence differences between aligned pairs of amplicons to delineate OTUs, using k-mer comparisons and a new and extremely fast global pairwise alignment algorithm. During the breaking phase, Swarm uses amplicon abundance information and OTUs’ internal structures to refine the clustering results.

In our view, amplicons can be seen as discrete coordinates in an abstract amplicon-space. Each position of this space represents a possible amplicon, which can be absent from the dataset (null abundance), or present (observed abundance). The direct neighbors of a given amplicon are all the possible amplicons with one nucleotide difference (a substitution, insertion, or deletion). That notion can be extended to d-neighbors, amplicons with d nucleotide differences. In that context, clusters are contiguous regions of the amplicon-space with non-null abundances. The assumption behind Swarm is that clusters are clearly separated by empty regions, i.e., amplicons do not form a vast continuum. If this condition holds true, then OTUs can be allowed to grow iteratively until they reach their natural limits (i.e., the empty regions of the amplicon-space).

Swarm explores the amplicon-space as follows: Swarm processes the input file and creates a pool of amplicons. An empty OTU is created, and the first available amplicon in the pool is withdrawn from the pool to become the OTU seed. The seed is then compared to all amplicons remaining in the pool, and the measured number of differences is stored in the memory. The number of differences is calculated as the number of nucleotide mismatches (substitution, insertion, or deletion) between two amplicons once the optimal pairwise global alignment has been found. Amplicons for which the number of differences is equal to or less than d, the user-chosen local clustering threshold, are removed from the pool and added to the OTU where they become subseeds.

Each subseed is then compared to the amplicons remaining in the pool, but only to those that have at most 2d differences with the seed. Indeed, amplicons with more than 2d differences with the seed cannot have d or less differences with one of its subseeds. This filtering step can be generalized as such: a subseed of generation g, a g-seed, is compared to the amplicons remaining in the pool, but only to those that have at most (g + 1) d differences with the seed.

After each series of comparisons, amplicons for which the number of differences is equal to or less than d are removed from the pool and added to the OTU to become (g + 1)-seeds. This iterative growth process is repeated for each generation of subseeds as long as new amplicons are captured. The OTU is then closed. The first available amplicon is removed from the pool, becomes the seed of a new OTU, and the process is repeated until no more amplicons remain in the pool.

This clustering process generates stable OTUs, regardless of the first seed choice. Thus, an OTU organically grows to its natural limits where it cannot recruit any more amplicons with d or fewer differences. Operating in this way, Swarm removes the two main sources of variability inherent in greedy de novo clustering methods: the need to designate an OTU center (centroid selection), and the need for an arbitrary global clustering threshold (maximum radius). Swarm outlines OTUs without imposing one particular shape or size, and produces the same OTUs regardless of the initially selected amplicon.

Swarm’s process guarantees that any amplicon in a given OTU has at least one neighbor with d or less differences. In other words, an OTU is a contiguous collection of amplicons (no internal gaps). Swarm also guarantees that OTUs are separated by gaps of at least d + 1 differences. Mechanically, it derives that using d = 1 will yield the finest partition of the amplicon-space, after the growth phase.

Under certain conditions, when using short or slowly evolving markers, or high d values for instance, the assumption that amplicons do not form a vast continuum can be violated. To solve this issue, Swarm implements a breaking phase that uses the structure of the cluster and amplicon abundance values to eliminate weak contiguity regions, and to delineate higher-resolution OTUs (see “Refining the clusters”).

Material and Methods

Swarm input files

Swarm input is a standard fasta file of unambiguous DNA or RNA amplicons, preferably fully sequenced (from the forward to the reverse primer), and with unique identifiers. The amplicon identifier is defined as the string comprised between the “>” and the first space or the end of the line, whichever come first. Swarm expects amplicon identifiers to be unique to avoid ambiguity in the clustering results. The amplicon sequence can only be composed of A, C, G and Ts (or Us) (case insensitive), and Swarm exits with an error message if any other symbol is present. To reduce the volume of data and to further increase analysis speed, it is recommended to merge amplicons with strictly identical sequences (dereplication). The Swarm README file gives examples of commands to properly format, and dereplicate the input fasta file.

Avoiding unnecessary pairwise global alignments

Global pairwise alignment is an exact but computationally expensive way to count the number of differences between two amplicons. It is possible to avoid costly comparisons by simply comparing amplicon lengths: two amplicons cannot have less than d differences if their length difference is greater than d. Unfortunately, that costless filtering is rather limited in scope. A more general and powerful way to estimate amplicon identity is to compare k-mer compositions (Ukkonen, 1992). Swarm starts by listing all possible DNA oligomers of length 5 (i.e., 5-mers; 45 = 1,024 possibilities). For a given amplicon, Swarm counts the number of occurrences of each 5-mer and builds a 1,024-bit vector of zeros (absence or even number of occurrences) and ones (odd number of occurrences). For d differences between two amplicon, the maximum number of differences in their k-mer vectors is 2dk. Therefore, if d = 1, two amplicons with one difference in their sequences will have at most 10 differences in their 5-mer vectors. Only pairs of amplicons with 10 or less differences in their 5-mer vectors need to be considered as candidates for a pairwise global alignment. It is possible for a pair of dissimilar amplicons to have similar bit vectors, but it is not possible for two similar amplicons to have dissimilar bit vectors. In other words, false positives are possible but not false negatives. As comparing bit vectors can be done rapidly in modern computers using XOR and POPCOUNT bitwise instructions, Swarm can efficiently, and in a lossless way, filter out many unnecessary pairwise global alignments.

Pairwise global alignment implementation

To speed up the remaining pairwise comparisons, Swarm implements a novel exact global pairwise alignment algorithm (Needleman & Wunsch, 1970) using SIMD vectorial instructions of modern CPUs, similar to the local pairwise alignment (Smith & Waterman, 1981; Gotoh, 1982) implemented in SWIPE by Rognes (2011). Swarm is able to compare up to 16 pool amplicons simultaneously with one seed (or k-seed) per CPU core. Swarm is also multi-threaded and efficiently uses multiple cores. In addition to computing the alignment score, it also stores backtracking data during amplicon comparisons in order to subsequently reconstruct the highest scoring alignments, and to count the number of differences. Since amplicons are usually of limited length (less than 1,000 bp) for the intended purpose, the amount of storage required for this data is not a major concern.

To achieve the most efficient parallelization of the algorithm, using as many parallel operations as possible (i.e., 16 simultaneous pairwise comparisons per CPU core), the magnitude of the score values of the highest scoring alignments to be calculated should be small, preferably fitting in one byte of memory (a byte can store one integer value ranging from 0 to 255). Swarm calculates the alignment score as follows: instead of computing the optimal global alignment similarity score as in the Needleman-Wunsch algorithm, Swarm identifies the alignments with the minimum edit distance, as described by Sellers (1974), by transforming the given similarity scoring system into an equivalent edit distance system. In the default scoring system a match is given a score of + 5, a mismatch −4, gap opening −12, and gap extension −4. Swarm will transform this scoring scheme into an equivalent system of positive integers where each mismatch corresponds to a penalty of + 9, gap-opening + 12, and gap extension + 7. This modified scoring system yields pairwise alignments strictly identical to the pairwise alignments produced by the original scoring system.

Swarm first tries to compute the global pairwise alignment using a single byte of memory, allowing scores up to 255, which corresponds to a maximum of 28 differences. Should the score be larger than 255, Swarm will re-compute the pairwise alignment using 2 bytes of memory or more. Using two bytes of memory reduces the speed of the pairwise alignment by 50%; however the upstream k-mer filtering implemented in Swarm limits the number of alignments overflowing one byte of memory.

After having identified the best alignment(s), Swarm will backtrack the optimal alignment (or one of the co-optimal alignments) using the backtracking data saved earlier. It will then simply count the number of mismatches and indels in that alignment and use it as the final difference between two amplicons.

In a very few cases the co-optimal alignments might contain a different number of gaps. As a result, the final number of differences counted may depend on which alignment is followed during the backtracking. The effect of this alignment issue is that a very small variability may exist in some results. We are currently solving this minor issue, and it will be implemented in a future Swarm release.

Refining the clusters

Swarm is an agglomerative, unsupervised, single-linkage-clustering algorithm. Single-linkage clustering is known to produce chains of amplicons that can potentially link closely related OTUs and decrease clustering resolution (Huse et al., 2010). As Swarm works with a small, user-chosen local clustering threshold value (d = 1 by default), this chaining effect is rather limited. Nevertheless, this risk exists, especially when using short or slowly evolving molecular markers. To solve this issue, we implemented an algorithm that explores the internal structure of the OTUs produced by Swarm, and detects and breaks possible chains by using abundance values associated with each amplicon.

OTUs present an internal structure where the most abundant amplicon usually occupies a central position and is surrounded by less abundant amplicons. The way Swarm explores the amplicon-space naturally produces a graph representation of the OTU, in the form of a star-shaped minimum spanning tree. In this context, chains of amplicons appear as links between such star-shaped sub-graphs. To identify and break the chains, our algorithm finds paths linking abundant amplicons (peaks) and monitors the abundance variations along these paths. If abundances decrease, go through a minimum, and then go up again (valley shape), this indicates a possible amplicon chain. Depending on the depth of the valley (i.e., the ratio between the minimum and maximum observed abundances), the algorithm will decide whether or not to break the graph into independent OTUs. In its current form, this step in Swarm is a python companion script set with the following parameters: only amplicons with 100 or more copies are considered as peaks; for a given pair of peaks, the path always starts from the highest peak (primary peak) and joins the secondary peak; valleys are considered deep when the ratio between the lowest point and the secondary peak is equal or greater than 50. Observations we made on environmental data indicated that chains of amplicons tend to form in the largest clusters, and are unlikely to occur in clusters with peaks smaller than 100. These smaller clusters are not removed from the clustering results, they are merely not searched for potential chains.

As this OTU breaking step improves Swarm’s precision at all clustering levels (see Figs. S2 and S3), we recommend always applying this companion script. We are currently testing a faster, less parametrized and more elegant solution using graph properties to quickly identify potential breaking points. We plan to include it soon in Swarm itself.

On the influence of the value d

Swarm’s most important user-chosen parameter is d, the local clustering threshold (for other parameters, see the Swarm Manual). Empirical results show that the choice of the d value has far less impact on Swarm’s clustering results than the choice of the global clustering threshold has on the results of other methods (see results below where we show that the choice of d has minimal impact). By default, d is set to 1 to obtain the finest partition of the amplicon-space, and to maximize the yield of biological information. Several factors can motivate the use of higher d values: longer amplicons, fast evolving markers, shallow sequencing (i.e., under-sampling of the amplicon-space), or the need to work with fewer but more inclusive OTUs. The stability of Swarm results also allows hierarchical clustering approaches: tracking OTU coalescing events when d increases allows to identify the d value best suited for the targeted amplicon-space region. We invite users to test several values to find the d value, or the range of d values, best fitting their data and scientific questions.

Speed and general behavior

Swarm clustering is a non-linear process. For speed purposes, it is suggested to sort the input data by decreasing amplicon abundance (if applicable); however results will not change, only the speed. Abundant amplicons often occupy a central position in OTUs, and when used as seeds, rapidly capture large amounts of subseeds. Using this strategy, the first OTUs produced are generally among the largest. As OTUs are outputted as soon as they are closed, most of the amplicons can be quickly removed from the pool early in the clustering process, and can be passed to post-clustering analyses. The duration of a Swarm analysis depends on several parameters: e.g., the d value used, the amplicon number and length, the number of CPU cores available, the molecular diversity of the dataset, and the amplicon abundance distribution. While being slower than Usearch, the fastest greedy heuristic for amplicon clustering, Swarm is fast enough to deal with datasets with millions of unique amplicons in a few days. For example, the clustering of an unpublished BioMarKs (http://biomarks.eu/) dataset of 1.5 million raw reads (312,503 unique amplicons, 382 nucleotides on average) takes less than 7 or 6 min (d = 1 or d = 7, respectively) using a single thread, while about 3 min (d = 1 or d = 7) using 16 threads. These timing experiments were performed with Swarm version 1.2.12 on a machine running Linux 2.6 with dual Intel Xeon E5-2670 processors (2.6 GHz) having a total of 16 physical cores and 64 GB RAM.

Swarm output files

The dereplication command example provided in Swarm’s README includes amplicon copy number within the amplicon identifier. This amplicon copy number information is retained in the output file, and used by Swarm to output statistics on the number of unique amplicons in the swarm, total copy number of amplicons in the swarm, identifier of the initial seed, initial seed abundance, number of singletons (amplicons with an abundance of 1), maximum number of iterations, the maximum radius of the swarm.

Swarm default behavior is to output results using a format similar to DNAclust’s format (but with space separated amplicon identifiers instead of tab-separated). An optional behavior is to output results using the Usearch format, which allow easy integration with extant amplicon-analysis pipelines, such as QIIME (Caporaso et al., 2010). Swarm can also optionally output results in a format suitable for easy integration with Mothur (Schloss et al., 2009).

Mock-communities

As Swarm was designed to analyze extremely large high-throughput sequencing datasets, we compared it to other stand-alone, fast, de novo clustering methods. Other clustering methods, such as average linkage, complete linkage or hierarchical clustering, do not scale up to these large datasets, and were therefore not compared.

Swarm’s performance was tested on two mock communities each comprising the same genome isolates from 49 bacterial and 10 archaeal species: one with even populations yielding 143,163 unique 16S amplicons (1,576,869 raw reads of average length 254.6 bp), the other with uneven populations yielding 55,622 unique 16S amplicons (637,693 raw reads of average length 253.9 bp). These 59 genomes derived from 57 species, as two strains of Shewanella baltica and two strains of Methanococcus maripaludis were used (See Table S1 for a complete list). For the clustering algorithm comparisons, these strains were collated as we only used the species-level taxonomic classifications. To construct the uneven mock community, genome isolates were distributed according to a log-normal distribution with parameters fitted from a soil microbial community. This gave a more realistic community structure with a few abundant species but a majority of rare organisms. In both cases, sequencing on the MiSeq platform with 250 bp paired-end reads was performed following amplification of the V4 region of the 16S rRNA gene with fusion Golay adaptors barcoded on the reverse end. The forward 16S rRNA primer sequence 515f was used (GTGNCAGCMGCCGCGGTAA). The reverse primers, barcodes and adaptors were identical to Caporaso et al. (2011).

Comparison with popular de novo clustering methods

Sequences were trimmed for quality and adaptors using Sickle (https://github.com/najoshi/sickle) and then forward and reverse reads were overlapped with PandaSeq (Masella et al., 2012) insisting on a 50 bp overlap. Strictly identical amplicons were merged with the command described in Swarm’s README, and the two datasets were subjected to five different stand-alone clustering methods: CD-HIT-454 (Fu et al., 2012, v4.6 2012-11-12), DNAclust (Ghodsi, Liu & Pop, 2011, v64bit, release_3, 2013-08-12), Swarm (v1.2.3) with OTU breaking, Usearch (Edgar, 2010, v7.0.1001_i86linux32) with presorting of the amplicons by decreasing length (option -cluster_fast) and without presorting (options -usersort -cluster_smallmem). Different clustering thresholds were used: d = 1–20 local differences for Swarm, and t = 1–20% global divergence for the other methods. For each clustering threshold and each clustering method, the first analysis was done on a fasta file sorted by decreasing abundance, and then repeated 100 times with amplicon input order randomly shuffled.

The clustering results of the two communities were then evaluated with three metrics using the known assignments to the 59 species as the ground truth, which was determined by matching with Usearch against the known 16S rRNA reference sequences and only using reads that were within 5% sequence difference using Usearch with the usearch_global alignment option. The three metrics used were: recall, which quantifies the extent to which amplicons assigned to the same species are grouped together in the same OTU (i.e., not over-splitting); precision, which asks if all amplicons in an OTU are assigned to the same species (i.e., not over-grouping); and the adjusted Rand index, which summarizes both precision and recall as the proportion of pairs of amplicons that are placed in the same OTU and are from the same species, but adjusting for the expected proportions through random chance (Rand, 1971; Hubert & Arabie, 1985). The results were synthesized in Fig. 2 (uneven community) and Fig. S1 (even community) using R (R Development Core Team, 2014) and the ggplot2 library (Wickham, 2009). The command lines and scripts we used to analyze the data, and to visualize the results are provided in the File S1.

Figure 2 Uneven mock-community.

Comparisons of five clustering methods, over 20 different clustering thresholds, and 100 amplicon input-order shufflings of a community composed of species of uneven abundances. Clustering precision and recall are estimated using amplicon taxonomic assignments as ground truth, and are summarized by the adjusted Rand index.

Results and Discussion

For both uneven (Fig. 2), and even communities (Fig. S1), almost all clustering thresholds for CD-HIT-454 and Usearch (with or without presorting) show extreme variability due to strong input-order dependency, while Swarm and DNAclust do not. Because of the pervasive variability in other methods, we then compared the median output values (see Table S1). Using the adjusted Rand index for both the uneven and even populations, Swarm outperforms the other methods at small d or t values (1–2); these small clustering thresholds are of critical importance when making fine-scale partitions of amplicon datasets, and strongly indicate that Swarm is more robust than the other methods to sequencing noise. At medium values (3–5), Swarm is either better, tied, or within 0.011 points to DNAclust. Swarm outperforms all other methods at larger values (6–20). These superior results for Swarm reflect its ability to conserve good precision and good recall over a wide range of clustering thresholds, which is critical as the true threshold for species-level assignments will in general not be known in advance and will vary with the choice of the sequenced marker or genomic region. Additionally, the adjusted Rand index results show that Swarm results are little affected by the choice of d; that is, Swarm limits the effect of the choice of clustering threshold and is adaptative across a large array of organisms and genes.

Swarm is efficient enough to deal with today’s largest datasets, and several new optimizations are now in development to handle even larger future datasets. For example, with the 256-bit integer SIMD instructions of the new Intel Haswell CPUs (released in late 2013) Swarm’s pairwise alignment throughput can double. We are also testing more efficient parallelization strategies, as well as new filters to avoid further un-needed pairwise alignments. These hardware evolutions and software optimizations will improve Swarm’s scalability even further. In parallel, improvements to our abundance-based chain breaking model will increase Swarm’s capacity to produce robust OTUs and meaningful biological results.

In summary, Swarm is a novel and robust approach that solves the problems of arbitrary global clustering thresholds and centroid selection induced input-order dependency, and creates robust and more natural OTUs than current greedy, de novo, scalable clustering algorithms. Swarm is a scalable C++ program able to handle many millions of amplicons. It is freely available at https://github.com/torognes/swarm under the GNU Affero General Public License version 3.

Supplemental Information

Supplemental Information 1 LaTeX source files

Click here for additional data file.

Table S1 Biological composition of the even and uneven mock-communities used for this study

Click here for additional data file.

Table S2 Uneven mock-community. Median values for Fig. 2

Click here for additional data file.

Table S3 Even mock-community. Median values for Fig. S1

Click here for additional data file.

Figure S1 Even mock-community

Comparisons of five clustering methods, over 20 different clustering thresholds, and 100 amplicon input-order shufflings of a community composed of species of even abundances. Clustering precision and recall are estimated using amplicon taxonomic assignments as ground truth, and are summarized by the adjusted Rand index.

Click here for additional data file.

Figure S2 Uneven mock-community, without swarm’s OTU breaking

Comparisons of five clustering methods, over 20 different clustering thresholds, and 100 amplicon input-order shufflings of a community composed of species of even abundances. Clustering precision and recall are estimated using amplicon taxonomic assignments as ground truth, and are summarized by the adjusted Rand index.

Click here for additional data file.

Figure S3 Even mock-community, without swarm’s OTU breaking

Comparisons of five clustering methods, over 20 different clustering thresholds, and 100 amplicon input-order shufflings of a community composed of species of even abundances. Clustering precision and recall are estimated using amplicon taxonomic assignments as ground truth, and are summarized by the adjusted Rand index.

Click here for additional data file.

File S1 Code and commands used to perform the analyses

Click here for additional data file.

We would like to thank Umer Ijaz for help in running analyses. We are grateful to the CNRS-UPMC ABiMS bioinformatics platform (http://abims.sb-roscoff.fr/) and to the Regional Computing Center at the University of Kaiserslautern for providing computational resources and support.

Additional Information and Declarations

Competing Interests

Author Contributions

The authors declare there are no competing interests.

Frédéric Mahé conceived and designed the experiments, performed the experiments, analyzed the data, wrote the paper, prepared figures and/or tables, reviewed drafts of the paper.

Torbjørn Rognes conceived and designed the experiments, performed the experiments, wrote the paper, reviewed drafts of the paper.

Christopher Quince analyzed the data, contributed reagents/materials/analysis tools, wrote the paper, prepared figures and/or tables, reviewed drafts of the paper.

Colomban de Vargas conceived and designed the experiments, wrote the paper, reviewed drafts of the paper, Dr. de Vargas provided useful advice and feedback.

Micah Dunthorn conceived and designed the experiments, analyzed the data, wrote the paper, prepared figures and/or tables, reviewed drafts of the paper.

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
