# Peer review of "Swarm: robust and fast clustering method for amplicon-based studies"

_PeerJ, doi:10.7717/peerj.593_

## Round 0.1 · original submission · Minor Revisions

· Academic Editor

Minor Revisions

I have now received reviews from two experts on clustering methods. Here are what I view as the principal concerns raised by the reviewers.

The testing of Swarm with the mock community does not seem sufficient for two reasons. First, it is not clear that clustering the organisms into the named species of bacterial taxonomy is a good criterion for judging the quality of the program. I will add to what Jason Wood has said on this by noting that the species of taxonomy are known to be highly diverse in their ecology, and indeed usually contain multiple ecologically distinct clusters that I think should have been picked up by Swarm. I’ll also add that a recent paper judging the goodness of different algorithms aimed to find “ecological consistency” of the clusters found by various algorithms (http://www.ploscompbiol.org/article/info%3Adoi%2F10.1371%2Fjournal.pcbi.1003594 ; Schmidt et al. 2014. Ecological Consistency of SSU rRNA-Based Operational Taxonomic Units at a Global Scale). Finding *taxonomic* consistency doesn’t seem as good an aim, so please justify and explain better your choice of criterion. Secondly, you haven’t provided any substantive details about the organisms that were chosen and how they were distributed.

Wood notes that you haven’t explained why you believe that the chain-breaking algorithm is improving precision at all levels.

The anonymous reviewer notes that your method of refining the clusters does not seem to scale with sample size.

The anonymous reviewer has asked for more details about timing results; the reviewer also suggests ways to re-format the paper more in keeping with PeerJ style.

I would like to add a couple of important points. First, you discuss how the algorithm will allow OTU’s to reach their “natural limits,” but it is not clear what you mean by that. Also, some recent investigators (including the Schmidt et al. paper above) have shown that complete linkage clustering tends to give more ecologically consistent clusters than single-linkage clustering; so these are additional points you should consider.

I’d like to sum up by noting that both reviewers and I see the value of your paper in introducing an algorithm with important advantages over other clustering methods, but that there are points that should be addressed. If you would like to further pursue publication in PeerJ, would you please address the points I have outlined here, plus all the other points addressed by the reviewers?

Reviewer 1 ·

Basic reporting

The structure of the paper does not fully adhere to any of the templates provided. This is easily fixed by some renaming and rearranging of the sections. For example:

"Swarm's rationale" could be a subsection of the Introduction.
"Swarm's mechanics" could become Materials and Methods.
"Comparison with ..." could be split between Materials and Methods and Results and Discussion.
"Perspectives" could become Conclusions.

PeerJ policy states that Acknowledgements "should not be used to acknowledge funders -that information will appear in a separate Funding Statement on the published paper."

Experimental design

Some information is missing in order to make the paper reproducible.

Under speed and general behavior, timing results are mentioned but a description of the machine used to obtain these results is not. This information is also not provided when describing the comparison to other methods. Were all experiments performed on the same machine? What was it? It also would be of interest to provide timing results for each algorithm.

One assumes that the "even" mock community has equal abundance for each genome. Please provide more details as to the composition of the "uneven" mock community. How exactly are the abundances distributed?

The method used for "refining the clusters" depends upon somewhat arbitrary constants 50 and 100. Some explanation of how these were derived would be useful. Perhaps some indication as to how they might scale with the size of the data set. Also, it is not explicitly stated in the "Comparison" section whether on not the version of Swarm used included running the refinement companion script (though it is implied). It would be interesting to see the results before the script is applied.

Validity of the findings

No comments.

·

Basic reporting

Beyond some minor problems detailed in the general comments section, the article is clearly written and appears to satisfy the requirements of the journal.

Experimental design

While not important enough to hold up publication, I would have liked to see how well Swarm performs with environmental data in addition to the two mock communities.

Validity of the findings

No Comments. Everything looks good.

Additional comments

This paper by Mahé et al. describes a new program available for clustering large databases of amplicons that does not suffer from problems related to the ordering of the input sequence data or from arbitrary global clustering thresholds. The clustering of sequence data into OTUs is an important exercise for the understanding of microbial diversity in a community. Since currently available methods suffer from global clustering thresholds and the ordering of sequence data, Swarm could potentially serve as a much needed refinement of method for microbiologists.

The authors demonstrate the fidelity of their new program by constructing artificial communities using "genome isolates from 49 bacterial and 10 archaeal species", but fail to specify what they consider to be a 'species' or provide enough information to guess at the definition used (no list of organisms is provided). Are the authors using so-called 'named-species' (ie strains of Escherichia coli), or are they using a definition more like the ecological species concept (aka ecotype)? Regardless, Swarm appears to have much higher fidelity than other programs available.



Specific comments (line numbers from main.tex):

lines 141-145: "This filtering step can be generalized as such ..."
The variable 'k' is used twice in the text, for k-seed and k-mer. Since k-mer is frequently used for an oligomer of length k, I recommend changing the 'k' in k-seed to something else to help clarify.

lines 182-184: "The amplicon sequence ..."
The README file (see linearization) on github mentions the need for sequence data to be on a single line. It is unclear whether this is a requirement of Swarm, or just a recommendation.

lines 203-205: "For d differences between ..."
The variable 'k' in k-mer is assumed to be used in the formula 2dk, but it is not as clear as it could be (see note about lines 141-145).

lines 311-313: "This small clusters are not removed ..."
This should read: "Small clusters are not removed ..."

lines 315-316: "As this OTU breaking step improves Swarm's precision at all clustering levels ..."
No evidence or reference provided for this statement.

lines 386-388: "Other clustering methods ..."
The phrase "do not scale up to this large datasets" should probably read "do not scale up to these large datasets".

lines 390-395: "Swarm's performance was tested on two mock communities ..."
Who are the 49 bacterial and 10 archaeal species in the mock communities? I would like to see this in Supplementary Information if not in the text itself.

Table 1 and 2:
These appear to provide the same data as the box plots in Figure 1 and 2. I suggest moving these tables to Supplementary Information.


Jason M. Wood

---

## Round 0.2 · Minor Revisions

· Academic Editor

Minor Revisions

Thank you for your substantial improvements in the clarity of your paper. I still have two issues that need to be resolved: one is a matter of clarity and the other could possibly require a substantial amount of work, depending on what you actually did in constructing your mock community.

First, I'd like you to explain "d" better in the rationale section. You need to make clear that this parameter is user-supplied, and you need to explain what it means. For example, if I understand "d" correctly, you should explain more clearly that a particular value of d will allow discovery of clusters that can be bridged only by a divergence level greater than d. In other words, you should explain what are the consequences of a particular value of d. Also, does d evolve during the process? It would probably help to include a little bit of foreshadowing here to state that d=1 gives clusters that best match the taxonomic clusters.

Related to my uncertainty around d, I was confused by your statement in "On the influence of the value of d" that "the d value has far less impact than the choice of the global clustering threshold." This was confusing because you have touted Swarm for not requiring a global threshold, so I assume you meant that the d value in Swarm has less effect than the global clustering thresholds of other algorithms. Please make this clear.

Now, here is the critical issue that I am the most concerned about. I appreciate your providing Supp. Table 1, but it is still not clear to me how many organisms you included in your mock communities and how much diversity you included within each of the species you provided. If you simply took one or two organisms from each species taxon, and were not careful to include species that were closest relatives, then it seems you have not adequately tested your algorithm's ability to conduct de novo clustering. If what you have done is to mostly supply one organism from each species, and you have not been careful about picking most-closely-related species pairs, then I urge you to re-do the tests of the algorithms on a more challenging and realistic mock community set. You can't claim that you're finding the species when you're not providing any within-species diversity or most-closely-related species.

If you are able to address these issues, I'd look forward to reading your next version of the paper.

---

## Author Rebuttal · Round 0.2

Dr. Frédéric Mahé
Technische Universität
Erwin-Schrödinger Str. 14
67663 Kaiserslautern
Germany

Dr. Frédéric Mahé, Technische Universität, Erwin-Schrödinger Str. 14, 67663 Kaiserslautern, Germany

Dr. Frederick Cohan,
Academic Editor for PeerJ

July 22, 2014

Dear Sir,

We would like to thank you for considering our manuscript for review. We modified the manuscript according to the comments and suggestions of the reviewers. We are grateful to them for highlighting some important points, and we believe that this has improved the manuscript. Our responses to the reviewer's comments are detailed below.

We hope that these modifications and responses address the reviewers' comments sufficiently and make our manuscript acceptable for publication in PeerJ.

Sincerely,

Frédéric Mahé and co-authors
* * *
### Editor's comments

**I have now received reviews from two experts on clustering methods. Here are what I view as the principal concerns raised by the reviewers.**

**The testing of Swarm with the mock community does not seem sufficient for two reasons. First, it is not clear that clustering the organisms into the named species of bacterial taxonomy is a good criterion for judging the quality of the program.**

**I will add to what Jason Wood has said on this by noting that the species of taxonomy are known to be highly diverse in their ecology, and indeed usually contain multiple ecologically distinct clusters that I think should have been picked up by Swarm. I'll also add that a recent paper judging the goodness of different algorithms aimed to find "ecological consistency" of the clusters found by various algorithms (Schmidt et al. 2014, *Ecological Consistency of SSU rRNA-Based Operational Taxonomic Units at a Global Scale*). Finding taxonomic consistency doesn't seem as good an aim, so please justify and explain better your choice of criterion. Secondly, you haven't provided any substantive details about the organisms that were chosen and how they were distributed.**

Several points are raised here.

Regarding the use of taxonomic assignment as ground truth. Most users of clustering programs want results that do fit best with traditional taxonomic classifications, and those classifications do reflect years of expert knowledge and real species differences. We agree that there are problems with the bacterial taxonomy that make bacterial species an imperfect standard to compare to but there is no better alternative that we know of and these imperfections apply to all the algorithms equally since the same test data set was used, hence, these species provide a valid benchmark for comparison across algorithms.

The findings reported in Schmidt et al. (2014) *Ecological Consistency of SSU rRNA-Based Operational Taxonomic Units at a Global Scale* are of great interest, but their recommendation to use hierarchical complete linkage clustering as the default choice for OTU clustering is impractical. Swarm and the other clustering methods tested in our manuscript are designed to work with the large datasets generated today. Hierarchical complete linkage clustering requires all-vs-all amplicon-distance computation, a task that very rapidly exhausts available computation and memory resources. Schmidt et al. approach also requires an additional layer of information, namely the keywords describing the sampled environments or physico-chemical parameter values. In our experience, these metadata often lack consistency, despite sample collectors best efforts.

Nonetheless, the objective of Schmidt et al. to get a finner-scale vision of molecular diversity beyond the traditional 97% veil is also ours. As noticed by the editor, Swarm was designed to detect fine-scale variability and can discriminate OTUs with as low as two differences, sometimes to the detriment of its "recall" score when these OTUs are under the same taxonomic assignment. Mixing the scalability and high-resolution of Swarm with the approach of Schmidt et al. to characterize the observed molecular diversity (is it linked to ecological diversity?) is a very promising idea, with the potential to vastly improve our understanding of ecosystems.

The list of species (49 bacterial and 10 archaeal strains) used to construct the even and uneven mock-communities is now available as a Supplementary Table 1.

**Wood notes that you haven't explained why you believe that the chain-breaking algorithm is improving precision at all levels.**

We added Supplementary Figures 2 and 3 to illustrate the impact of OTU breaking on Swarm clustering metrics.

**The anonymous reviewer notes that your method of refining the clusters does not seem to scale with sample size.**

The reviewer is right. Although, the speed of or OTU breaking step is not linked directly to the sample size, but to the number of highly abundant amplicons inside the OTU that is processed. The OTU breaking step gently slows down as the sample size increases, and becomes a computation bottleneck only for the largest OTUs we had to work with (TARA OCEANS and Earth Microbiome Project for example, with OTUs containing hundreds of thousand of unique amplicons). As mentioned in the manuscript, we developed a more elegant and scalable solution (available at https://github.com/larsmew/swarm_breaker2). Before implementing that new solution into swarm, some tests on larger environmental datasets are required, but we encountered a scalability problem with the script computing clustering metrics and we have to solve that problem first (see below).

**The anonymous reviewer has asked for more details about timing results; the reviewer also suggests ways to re-format the paper more in keeping with PeerJ style.**

We added details regarding the timings results, and the section titles were renamed to match PeerJ style.

**I would like to add a couple of important points. First, you discuss how the algorithm will allow OTU's to reach their "natural limits", but it is not clear what you mean by that. Also, some recent investigators (including the Schmidt et al. paper above) have shown that complete linkage clustering tends to give more ecologically consistent clusters than single-linkage clustering; so these are additional points you should consider.**

The natural limits of the clusters are the empty regions of the amplicon-space. That sentence have been clarified in the manuscript. As stated above and in the manuscript, hierarchical complete linkage clustering is known to give very good clustering results. Unfortunately, it would require unrealistic computation resources to apply hierarchical complete linkage clustering to present day amplicon datasets.
* * *
# Reviewer comments

**Reviewer 1**

**Basic reporting**

**The structure of the paper does not fully adhere to any of the templates provided. This is easily fixed by some renaming and rearranging of the sections. For example:**
**"Swarm's rationale" could be a subsection of the Introduction.**
**"Swarm's mechanics" could become Materials and Methods.**
**"Comparison with ..." could be split between Materials and Methods and Results and Discussion.**
**"Perspectives" could become Conclusions.**

The sections of the manuscript have been renamed as suggested.

**PeerJ policy states that Acknowledgments "should not be used to acknowledge funders—that information will appear in a separate Funding Statement on the published paper."**

Funding sources are now in the proper section.

**Experimental design**

**Some information is missing in order to make the paper reproducible.**

**Under speed and general behavior, timing results are mentioned but a description of the machine used to obtain these results is not. This information is also not provided when describing the comparison to other methods. Were all experiments performed on the same machine? What was it? It also would be of interest to provide timing results for each algorithm.**

Details on the type of computer used to get swarm timings and clustering results were added to that section. The timing results for each algorithm vary widely according to the number of amplicon, their length and their relative order. As a rule of thumb, usearch is the fastest algorithm, uclust and swarm are close from one another, cd-hit-454 is the slowest algorithm.

**One assumes that the "even" mock community has equal abundance for each genome. Please provide more details as to the composition of the "uneven" mock community. How exactly are the abundances distributed?**

We used the same genomes to construct an uneven mock community where organisms were distributed according to a log-normal distribution with parameters fitted from a soil microbial community. This gave a more realistic community structure with a few abundant species but a majority of rare organisms.

**The method used for "refining the clusters" depends upon somewhat arbitrary constants 50 and 100. Some explanation of how these were derived would be useful. Perhaps some indication as to how they might scale with the size of the data set. Also, it is not explicitly stated in the "Comparison" section whether on not the version of Swarm used included running the refinement companion script (though it is implied). It would be interesting to see the results before the script is applied.**

We clarified the "Comparison" section by stating that we used swarm and the OTU breaking script. We also added two supplementary figures (Supplementary Figures 2 and 3) showing swarm results **without** OTU breaking.

**The parameters used in the OTU breaking script were tested on a variety of 454 datasets, with different amplicon lengths. Empirical tests and visual inspections of the OTUs indicated that these parameters gave good results in general. These parameters were not finely tuned, and in fact can vary up to 10-fold without a significant impact on clustering metrics. As stated above, we have a more elegant and scalable solution that should be implemented in Swarm soon.**

As suggested, Supplementary Figures 2 and 3 now show Swarm clustering metrics without OTU breaking.
* * *
**Reviewer 2 (Jason Wood)**

**Experimental design**

**While not important enough to hold up publication, I would have liked to see how well Swarm performs with environmental data in addition to the two mock communities.**

That point is important to us too and we tried to apply the same clustering metrics pipeline to larger environmental datasets. Unfortunately, we encountered a scalability problem with our script computing the different clustering metrics. Despite our best efforts, we were not able to fix that problem during the revision window. To definitively solve that, we plan to completely rewrite the incriminated script in the coming weeks.

**Comments for the author**

**This paper by Mahé et al. describes a new program available for clustering large databases of amplicons that does not suffer from problems related to the ordering of the input sequence data or from arbitrary global clustering thresholds. The clustering of sequence data into OTUs is an important exercise for the understanding of microbial diversity in a community. Since currently available methods suffer**

**from global clustering thresholds and the ordering of sequence data, Swarm could potentially serve as a much needed refinement of method for microbiologists.**

**The authors demonstrate the fidelity of their new program by constructing artificial communities using "genome isolates from 49 bacterial and 10 archaeal species", but fail to specify what they consider to be a 'species' or provide enough information to guess at the definition used (no list of organisms is provided). Are the authors using so-called 'named-species' (ie strains of Escherichia coli), or are they using a definition more like the ecological species concept (aka ecotype)? Regardless, Swarm appears to have much higher fidelity than other programs available.**

The list of species (49 bacterial and 10 archaeal strains) used to construct the even and uneven mock-communities is now available as a Supplementary Table 1.

**Specific comments (line numbers from main.tex):**

**lines 141-145: "This filtering step can be generalized as such ..." The variable $k$ is used twice in the text, for $k$-seed and $k$-mer. Since $k$-mer is frequently used for an oligomer of length $k$, I recommend changing the $k$ in $k$-seed to something else to help clarify.**

As suggested, $k$ was replaced with $g$, to represent the number of "generations" (i.e. the number of growth iterations).

**lines 182-184: "The amplicon sequence ..." The README file (see linearization) on github mentions the need for sequence data to be on a single line. It is unclear whether this is a requirement of Swarm, or just a recommendation.**

It is only a recommendation. The README file have been clarified.

**lines 203-205: "For d differences between ..." The variable $k$ in $k$-mer is assumed to be used in the formula $2dk$, but it is not as clear as it could be (see note about lines 141-145).**

The ambiguity on the $k$ notation has been removed, and the formula should now read more clearly.

**lines 311-313: "This small clusters are not removed ..." This should read: "Small clusters are not removed ..."**

The sentence has been corrected as suggested.

**lines 315-316: "As this OTU breaking step improves Swarm's precision at all clustering levels ..." No evidence or reference provided for this statement.**

As stated above, we added two supplementary figures (Supplementary Figures 2 and 3) showing swarm results **without** OTU breaking to back up our statement.

**lines 386-388: "Other clustering methods ..." The phrase "do not scale up to this large datasets" should probably read "do not scale up to these large datasets".**

The sentence has been corrected as suggested.

**lines 390-395: "Swarm's performance was tested on two mock communities ..." Who are the 49 bacterial and 10 archaeal species in the mock communities? I would like to see this in Supplementary Information if not in the text itself.**

We used 59 bacterial and archaeal genome isolates from known cultured organisms. A genome list has now been added to the supplementary materials (Supplementary Table 1). These 59 genomes derived from 57 species, two strains of *Shewanella baltica* and two strains of *Methanococcus maripaludis* were used. For the algorithm comparisons we collated these strains only using the species level classifications.

**Table 1 and 2: These appear to provide the same data as the box plots in Figure 1 and 2. I suggest moving these tables to Supplementary Information.**

Table 1 and 2 were moved to Supplementary Tables.

––––––––––––––––––––––

---

## Round 0.3 · accepted · Accept

· Academic Editor

Accept

You've done a very nice job of clarifying the manuscript. I'd like to suggest that in the near future you might supply further documentation of Swarm's "recall" ability on Swarm's web site, by supplying more instances of species with multiple sequence entries in your mock communities.